# Prevention of Metabolic Syndrome by Phytochemicals and Vitamin D

**DOI:** 10.3390/ijms24032627

**Published:** 2023-01-30

**Authors:** Kazuki Santa, Yoshio Kumazawa, Isao Nagaoka

**Affiliations:** 1Department of Biotechnology, Tokyo College of Biotechnology, Tokyo 114-0032, Japan; 2Vino Science Japan Inc., Kanagawa 210-0855, Japan; 3Department of Biochemistry and Systems Biomedicine, Juntendo University Graduate School of Medicine, Tokyo 113-8421, Japan; 4Faculty of Medical Science, Juntendo University, Chiba 279-0013, Japan

**Keywords:** metabolic syndrome, phytochemicals, vitamin D, microbiota, TNF-α, adiponectin

## Abstract

In recent years, attention has focused on the roles of phytochemicals in fruits and vegetables in maintaining and improving the intestinal environment and preventing metabolic syndrome. A high-fat and high-sugar diet, lack of exercise, and excess energy accumulation in the body can cause metabolic syndrome and induce obesity, diabetes, and disorders of the circulatory system and liver. Therefore, the prevention of metabolic syndrome is important. The current review shows that the simultaneous intake of phytochemicals contained in citruses and grapes together with vitamin D improves the state of gut microbiota and immunity, preventing metabolic syndrome and related diseases. Phytochemicals contained in citruses include polyphenols such as hesperidin, rutin, and naringin; those in grapes include quercetin, procyanidin, and oleanolic acid. The intake of these phytochemicals and vitamin D, along with prebiotics and probiotics, nurture good gut microbiota. In general, *Firmicutes* are obese-prone gut microbiota and *Bacteroidetes* are lean-prone gut microbiota; good gut microbiota nurture regulatory T cells, which suppress inflammatory responses and upregulate immunity. Maintaining good gut microbiota suppresses TNF-α, an inflammatory cytokine that is also considered to be a pathogenic contributor adipokine, and prevents chronic inflammation, thereby helping to prevent metabolic syndrome. Maintaining good gut microbiota also enhances adiponectin, a protector adipokine that prevents metabolic syndrome. For the prevention of metabolic syndrome and the reduction of various disease risks, the intake of phytochemicals and vitamin D will be important for human health in the future.

## 1. Introduction

Excess energy accumulation from the intake of a high-fat, high-sugar diet and the typical Western diet causes metabolic syndrome. Generally, metabolic syndrome is defined as visceral obesity with two or more symptoms of hyperglycemia, hypertension, and dyslipidemia. A high caloric diet with a high-sugar and high-fat intake causes obesity, and such diets, when combined with a lack of exercise, become the first step toward metabolic syndrome [1,2]. As this status induces diabetes, cardiovascular diseases, and liver disorders, the prevention of metabolic syndrome is very important. In metabolic syndrome, the production of tumor necrosis factor α (TNF-α), a pathogenic contributor adipokine produced from enlarged adipocytes, induces chronic inflammation. This is a critical condition that in turn induces several disorders, which combine to cause metabolic syndrome [3]. On the other hand, the suppressed production of adiponectin, a protector adipokine, weakens the ability to overcome metabolic syndrome [4].

There are three types of adipocytes: white adipocytes, brown adipocytes, and beige adipocytes [5]. Healthy white adipocytes are important because they produce adiponectin and enhance the production of insulin from pancreatic β cells. During the progression of metabolic syndrome, however, adipocytes enlarge and change their shape to produce TNF-α. TNF-α suppresses the production of glucose transporter type 4 (GLUT4) and induces insulin resistance, which is responsible for chronic inflammation and causes diabetes [6]. Brown adipocytes come from white adipocytes. In adults, the browning of adipocytes is thought to be related to the control of body weight through the active metabolism of body fat [7]. Recently, attention has been focused on beige adipocytes [8]. Because beige adipocytes convert fatty acid and sugars into energy, which they then consume, increasing the proportion of beige adipocytes in the body has been suggested as being capable of preventing the onset of metabolic syndrome [9,10].

*Firmicutes* and *Bacteroidetes* are dominant types of gut microbiota, and each group of bacteria are classified into obese-type and lean-type [11]. *Firmicutes* are Gram-positive bacteria that are predominant in obese people. A high-fat, high-sugar diet, large amounts of meat consumption, and a diet containing few vegetables and fruits induce a *Firmicutes*-dominant gut microbiota. Some *Firmicutes* bacteria produce decomposing enzymes to digest indigestive fibers such as cellulose into glucose, potentially contributing to obesity [12]. On the other hand, a decrease in *Bacteroidetes*, which are Gram-negative bacteria, induces a thin mucin layer in the large intestine, weakens the cellular tight junctions, and causes leaky gut syndrome, hypertension, dyslipidemia, and lifestyle-related illness [13]. Therefore, lowering the F/B ratio is a good strategy for maintaining good health.

Consuming a healthy diet and exercising are important for preventing metabolic syndrome. The typical Western diet, with high fat and high sugar consumption, is responsible for metabolic syndrome. On the other hand, a Mediterranean diet, and the traditional Japanese diet, in which less fat, sugar, and meat are consumed, is less likely to cause metabolic syndrome [14]. In addition, the consumption of prebiotics, probiotics, polyphenols, and vitamin D, which is also known as 3PD, is a good strategy for nurturing good microbiota [15,16].

The phytochemicals in citruses and grapes prevent chronic inflammation by modifying good gut microbiota. Citrus phytochemicals include hesperidin, β-cryptoxanthin, rutin (quercetin-glycoside), and naringin; the phytochemicals in grapes include quercetin, procyanidin, and oleanolic acid (terpenoids) [17]. In addition, skin cells produce vitamin D from cholesterol after exposure to sunlight. Vitamin D was originally described as a bone hormone; in addition to these previously known effects, however, vitamin D is now considered to act as a type of steroid. Because large numbers of immune cells, including macrophages, express the vitamin D receptor (VDR), vitamin D is thought to participate in the regulation of immunity [18]. Furthermore, vitamin D not only binds to VDR and activates immune cells, but it also improves natural immune responses by increasing the production of antimicrobial peptides [19]. In addition, vitamin D is also an important substance for maintaining a diversity of good gut microbiota [20].

In metabolic syndrome, excess energy accumulation in the body arising from an imbalanced diet result in a variety of disorders such as diabetes, cardiovascular diseases, and Alzheimer’s disease. Instead, a healthy diet maintains good gut microbiota, suppresses chronic inflammation, upregulates immunity, and prevents metabolic syndrome. The current review explains how phytochemicals and vitamin D prevent metabolic syndrome and help to maintain healthy longevity.

## 2. Disorders Related to Metabolic Syndrome

Metabolic syndrome is a relatively new set of metabolic conditions/alterations that has been known since the early 21st century after the increase in obesity caused by changes in eating habits and lifestyles became apparent. It is characterized by various complications which are induced by diabetes accompanied by insulin resistance, such as hyperglycemia, arterial hypertension, and dyslipidemia. Furthermore, metabolic syndrome is a serious risk factor of early death because it causes cardiovascular, liver, and brain disorders or induces cancer through the presence of these disorders [21,22]. The main cause of metabolic syndrome is excessive energy accumulation in the body. An imbalanced diet with a high-fat and high-sugar intake and a lack of exercise are responsible for metabolic syndrome. These conditions cause dysbiosis in gut microbiota, with increased *Firmicutes* and decreased *Bacteroidetes* populations, induce the production of the pathogenic contributor TNF-α, and decrease the protective adiponectin, leading to the onset metabolic syndrome (Figure 1).

### 2.1. Diabetes

Diabetes is a part of metabolic syndrome and is classified into type 1 and type 2. In type 1 diabetes, damaged pancreatic β cells stop producing insulin; in type 2 diabetes, genetic factors related to inadequate insulin secretion are the main cause. Recently, the number of people with diabetes has been increasing in Japan. Two factors are related to the increase in metabolic syndrome: environmental factors including a high-fat diet, lack of exercise, obesity, visceral fat accumulation, and insulin resistance, and genetic factors including reduced insulin secretion. In addition, an unbalanced diet causes dysbiosis, induces insulin resistance, and leads to the onset of diabetes [23]. Blood sugar spikes, which are the state of temporary high blood sugar levels soon after meals, but which are not detected by conventional health checks because of their shortness, are of particular concern. This state is caused by the rapid decomposition of starches into glucose after meals by the secretion of insulin. Blood sugar spikes are huge health risks because they trigger arteriosclerosis-related disorders such as heart disease, stroke, and kidney malfunction. The intake of quercetin, a flavonoid with enzyme inhibitory properties, is becoming an attractive choice to stop blood sugar spikes and to prevent diabetes [24].

After the onset of obesity, adipocytes constantly produce free fatty acid (FFA). The continuous production of FFA affects macrophages, causing them to produce the proinflammatory TNF-α and to inhibit the glucose carrier GLUT4 synthesis in muscular cells. This inhibition leads to the onset of type 2 diabetes [25].

### 2.2. Cardiovascular Diseases

Glucose metabolism disorder, hypertension, and dyslipidemia induce the onset of atherosclerosis. Cholesterol is divided into two types: low-density lipoprotein (LDL) cholesterol and high-density lipoprotein (HDL) cholesterol. The LDL/HDL level is an indicator of arteriosclerosis. When this figure exceeds 2.5, the onset of arteriosclerosis is likely [26]. Cholesterol is an organic compound that is classified as a sterol subclass of steroids produced from acetyl-CoA, a metabolite of glucose and fatty acid. Acetyl-CoA is the substrate that produces ATP via the tricarboxylic acid cycle in mitochondria. However, the remaining substrate that is not used in ATP production is stored as cholesterol in the body. Cholesterol is used to produce a variety of hormones and bile acids. It is an essential material in biomembranes and an important substance for a variety of intracellular metabolic processes [27]. In general, HDL cholesterol is considered to be health-beneficial cholesterol because it acts as the carrier of accumulated cholesterol in peripheral tissue to the liver. On the other hand, LDL cholesterol transports excess cholesterol. However, oxidized LDL is incorporated into macrophages and is accumulated in vascular sub-endothelial spaces (intima), and it is responsible for the onset of arteriosclerosis [28]. Furthermore, a relationship between cellular senescence induced by chronic inflammation and arteriosclerosis has also been suggested [29].

Leaky gut, which is a state of increased intestinal vascular permeability caused by dysbiosis, induces chronic inflammation and triggers cardiovascular disorders. A high-fat and high-sugar diet causes dysbiosis and induces a deficiency of butyrate-producing bacteria, reducing regulatory T cells, weakening anti-inflammatory mechanisms, and increasing inflammation-inducible bacteria. As a result, the tight junction structure in the intestinal epithelium is weakened, and this state increases the permeability between cells in the intestine, resulting in the condition called leaky gut. This condition induces and influx of bacteria and lipopolysaccharides (LPS) from the lamina propria mucosae to the bloodstream and elicits inflammation by increasing cytokines and reactive oxide. The dysfunction of intestinal epithelial cells induces chronic inflammation and continuously provides factors that contribute to the pathophysiology of atherosclerosis, including cytokines, endotoxins, and gut microbes in the bloodstream [30,31]. Finally, TNF-α produced by chronic inflammation induces arteriosclerosis through foam cell formation from macrophages and moves into blood vessels [32]. Arteriosclerosis causes ischemic heart disease, cerebrovascular disease, and stroke. The prevalence of these disorders became remarkable after the Westernization of diets, including high fat and high sugar consumption. The decreased intake of vegetables and fruits, phytochemicals, and dietary fibers and the remarkable increase in the consumption of meats and dairy products are considered causes of the increased prevalence of arteriosclerosis.

### 2.3. Liver Disorders

Cholesterol in the liver moves to the intestine as a bile acid and is excreted from the body. Bile acid production is one of the main functions of the liver, with about 600 mL/day of bile acid being produced. The gallbladder stores bile acids and excretes them into the duodenum after food intake. Secreted bile acids work as an emulsification detergent and help the digestion of lipids by lipase. Bile acids exist as glycine/taurine conjugates and bile pigments, including bilirubin and biliverdin. Bilirubin is a metabolite which is produced from heme and originates from the heme protein in hemoglobin. Bile acid is excreted to the intestine and returns to the liver through enterohepatic circulation. However, deoxycholic acid, a second metabolite of bile acid produced by the gut microbiota, is harmful to the liver. Here, water-soluble dietary fibers absorb bile acids in the duodenum and incorporate cholesterol, then excrete them from the body through the large intestine [33]. Unbalanced dietary habits cause metabolic syndrome and dysbiosis, whereas a faulty gut microbiota increases the production of deoxycholic acid and ultimately induces liver injury [34,35]. Oleanolic acid, a phytochemical which is contained in grape skin, conjugates with the cholic acid receptor TGR5 as an agonist and prevents constipation by enhancing intestinal movement [36].

In addition, leaky gut syndrome causes non-alcoholic steatohepatitis (NASH). Increased permeability in the intestinal epithelium induces metabolic endotoxemia, insulin resistance, and inflammation and fibrosis in the liver through TNF-α production [37]. Unlike other types of hepatitis, such as alcoholic or viral hepatitis, NASH is induced by high fat and high sugar consumption, obesity, metabolic syndrome, and other lifestyle-related diseases. Obesity is considered a high-risk factor for liver cancer. The onset of non-alcoholic fatty liver disease (NAFLD) induces NASH and causes hepatic cirrhosis, and these liver disorders ultimately progress to cancer [38]. On the other hand, the liver has a high regenerative capacity. The improvement of dietary habits can ameliorate NASH-induced hepatic fibrosis [39], and the administration of quercetin metabolites protects the liver from acetaldehyde-induced cellular toxicity [40].

### 2.4. Brain Disorders

Recently, several studies have shown a relationship between metabolic syndrome and dementia [41]. In the United States, middle-aged people with obesity have a three-times higher risk of Alzheimer’s disease (AD) and a five-times higher risk of dementia induced by cerebral thrombosis, compared to other groups. AD has been called type 3 diabetes because it has the same pathophysiology as type 2 diabetes [42]. Neurotoxicity induced by diabetic brain inflammation and Tau protein peroxidation can lead to the onset of cognitive impairment. During AD progression, inflammation in microglial cells can be observed in the brain. A study conducted in AD patients showed that insulin administration through a trans-nasal pathway resulted in the direct transport of insulin to the brain. This experiment revealed that decreased levels of insulin in the brain induce AD, because low insulin levels in brain cells induce glucose deficiency and cell death [43].

In multiple sclerosis, proinflammatory Th17 cells that are activated by dysbiosis induce neural inflammation in the brain [44]. In contrast, short fatty acids produced by healthy gut microbiota increase tryptophan hydroxylase 1 (TPH1) expression in enterochromaffin cells and induce serotonin secretion in the gut. In addition, activated regulatory T cells (Treg) in the gut induce IgA-producing plasma B cells, which are producers of IL-10. Finally, these plasma B cells release IL-10 in the brain, thereby suppressing the inflammation associated with multiple sclerosis [45].

## 3. Citrus and Grape Phytochemicals and Vitamin D

### 3.1. Phytochemicals Contained in Citruses

Citruses contain a variety of phytochemicals. Japanese mandarin (orange) (*Citrus unshiu*) is one of the most popular fruits in Japan, along with apple and banana. Orange peel, known as *chenpi* in *Kampo* (Traditional Japanese) medicine, is often used in formulations for a variety of symptoms [46]. Citruses contain a variety of phytochemicals including hesperidin and β-cryptoxanthin. Rutin (quercetin-glycoside) is also found abundantly in Tartary buckwheat (*Fagopyrum tataricum*). Grapefruit, orange, and Hassaku orange contain naringin. *Citrus depressa* contains nobiletin [47]. Hesperidin, rutin, naringin, and nobiletin are flavonoids, and β-cryptoxanthin is a carotenoid found in citrus. Table 1 shows citrus phytochemicals and their effects in metabolic syndrome.

Hesperidin is abundant in mandarin oranges and sudachi (*Citrus sudachi*) and is effective for decreasing triglycerides, improving cold sensitivity, overcoming stress, improving skin condition, and recycling vitamin C. Hesperidin has a low absorption ratio in the body because its water solubility is low. Similarly, to other polyphenols, a glucose-conjugated form—hesperidin-glycoside—is widely used to improve bioavailability [48]. Hesperidin inhibits endotoxin shock in mice [49] and is effective for treating rheumatoid arthritis in humans [50]. In addition, hesperidin improves insulin resistance, hypertension, hyperglycemia, hypercholesterolemia, triglyceride, TNF-α, and hs-CPR levels in the blood [51,52].

**Table 1 ijms-24-02627-t001:** Anti-metabolic syndrome and anti-inflammatory effects of citrus phytochemicals.

Citrus Phytochemicals	Effects	Subjects	Ref
Hesperidin (flavonoids)	Endotoxin shock suppression	Mouse	[49]
	Alleviating rheumatoid arthritis	Human/Mouse	[50]
	Hyperglycemia, triglyceride, high blood pressure	Human	[51]
	Reduction of blood pressure, blood glucose, cholesterol, TNF-α, hs-CPR	Human	[52]
β-Cryptoxanthin (carotenoids)	Provitamin A effects: maintaining eyesight, helping growth and development	Human/Mouse	[53]
	Anti-stress effects by anti-oxidative effects	Human	[54]
	Bone homeostasis, osteoporosis prevention, bone metabolism	Human/ Mouse cells	[55]
	Effects for liver disorders (NFALD/NASH)	Human, etc.	[56]
	Metabolic syndrome and type 2 diabetes	Rat	[57]
	Reducing body fat levels, anti-oxidative stress response, prevention of ageing	*C. elegans*	[58]
Rutin (quercetin-glycoside: flavonoids)	Diabetes, blood glucose, anti-inflammatory effects, anti-oxidative effects	Human	[59]
	Alleviating arthritis	Rat	[60]
	Depletion of AGEs	Rat/Human cells	[61]
	Stress-induced injury, oxytocin receptor activation	Rat/Human cells	[62]
	Decreasing LDL, increasing HDL, improving learning capability	Rat	[63]
Naringin (flavanone-glycoside: flavonoids)	Enzyme activation related to tissue glucose intake from blood	Human/Rat/ Mouse/Cells	[64]
	Therapy of diabetes	Rabbit/Rat/ Mouse/Cells	[65]
	Suppression of LPS-induced TNF-α production	Mouse	[66]
	Anti-inflammatory effects in arthritis	Mouse	[67]
	Prevention of atherosclerosis	Mouse	[68]
	Improvement of circulatory system disease	Rat	[69]
Nobiletin (flavonoids)	Improving recognition, reducing soluble amyloid β	Mouse	[70]
	Enhancing circadian rhythms	Mouse	[71]
	Reducing the risk of metabolic syndrome	Human/Rat/ Mouse/Cells	[72]
	Alleviating metabolic dysregulation	Mouse	[73]

β-Cryptoxanthin, along with lycopene and astaxanthin, is a carotenoid. β-Cryptoxanthin is a provitamin A that is converted into vitamin A [53], has anti-oxidative effects, and can improve liver disease, arteriosclerosis, diabetes, and osteoporosis [54,55]. β-Cryptoxanthin is effective for preventing NFALD/NASH [56]; for suppressing hypertension, metabolic syndrome, and type 2 diabetes [57]; and reducing bodyweight and the oxidative stress response and preventing ageing [58].

Rutin was first found in the *Ruta graveolens* plant and has been used as a blood vessel protective agent. Rutin reportedly prevents diabetes, arthritis, chronic inflammation, and hay fever [59,60]. Other effects of rutin intake include the inhibition of the production of advanced glycation end products (AGEs) [61], oxytocin receptor activation [62], an increase in HDL-cholesterol, and a decrease in LDL-cholesterol and triglyceride [63].

Naringin is known to activate enzymes that catalyze glucose incorporation in tissues and to inhibit factors related to insulin resistance [64,65]. Naringin inhibits LPS-induced TNF-α production as well as hesperidin [66]. Other effects of naringin include inflammation suppression in arthritis [67], arteriosclerosis prevention [68], and cardiovascular disease prevention [69]. Nobiletin is effective for cognitive improvement in AD patients [70]. Furthermore, nobiletin is known to enhance circadian rhythms [71] and to reduce the risk of metabolic syndrome [72] and metabolic dysregulation in subjects with a high-fat diet [73].

### 3.2. Phytochemicals in Grapes

Phytochemicals found in grapes have made headlines due to their role in the French diet, as red wine is associated with a reduced incidence of heart disease [74]. According to some advocates, people in France tend to have lower cardiovascular disease rates than their American counterparts, and one of the reasons for this phenomenon is considered to be the consumption of polyphenols. The effects of resveratrol, which are contained in red wine, became a focus of media reports at one time, and resveratrol was suggested to have positive effects on extending life span [75]. Recent research has shown that resveratrol reduces the risk of obesity and NAFLD and improves the condition of the gut microbiota [76]. The polyphenols in red wine also reportedly prevent arteriosclerosis at the cellular level [77]. In addition, one glass of wine a day is part of the Mediterranean diet, which is generally considered to be a healthy diet [78].

The phytochemicals contained in grapes can be divided into three classes: terpenoids, carotenoids, and flavonoids [79]. Figure 2 shows these classifications of grape phytochemicals [80]. Grape flavonoids are widely used in functional foods and pharmaceutical products. Table 2 summarizes the anti-metabolic syndrome and anti-inflammatory effects of grape phytochemicals. Typical examples of grape phytochemicals for pharmaceutical and nutraceutical use are grape seed extracts (GSE) [81,82]. Resveratrol is another well-known grape phytochemical [83]. Recently, the anti-inflammatory effects of flavonoids, in addition to their anti-oxidative effects, have become a focus of interest. Flavonoids suppress chronic inflammation induced by TNF-α [84]. Recent research shows that flavonoids and procyanidin suppress the senescence-associated secretory phenotype (SASP), and their anti-ageing effects are of interest to many researchers [85,86].

### 3.3. Combination of Vitamin D and Phytochemicals

In general, skin cells synthesize vitamin D from cholesterol after exposure to UV light. However, the amounts of vitamin D in the human body tend to become inadequate due to a variety of reasons including differences in season and latitude as well as differences in the amount of melanin pigment contained in the skin [87]. The concentration of 25-OH-D3 measured in the blood is regarded as the blood vitamin D level. The intake of vitamin D from foods and supplements is encouraged because a blood vitamin D level of >30 mg/mL is regarded as sufficient. Vitamin D is a fat-soluble vitamin. Even though vitamin D is named as a type of vitamin, it is also considered a type of steroid hormone. Vitamin D is a bone-related hormone involved in the formation and decomposition of bones, and vitamin D deficiency impairs bone calcification, resulting in rickets in children and osteomalacia in adults [88,89]. Recently, research has been focused on the immune regulatory effects and anti-inflammatory effects of vitamin D. Vitamin D is subdivided into vitamin D2, which originates from plants, and vitamin D3, which originates from animals. Vitamin D3 works more efficiently in the human body [90]. Cholesterol is converted into vitamin D in skin cells under UV exposure and is further converted into 25-OH-D3 (calcifediol) by a liver enzyme; it then circulates in the blood after bonding with vitamin D-binding proteins. Enzymes in the kidneys and immune cells convert 25-OH-D3 into 1α, 25-(OH)2-D3. Together with retinoid acid, a metabolite of vitamin A, activated 1α, 25-(OH)2-D3 binds to vitamin D receptors (VDR) in macrophages. Finally, the vitamin D-VDR complex attaches to the promotor region of the TNF-α gene and stops the production of TNF-α [91].

Table 3 shows the anti-metabolic syndrome and anti-inflammatory effects of vitamin D. The intake of vitamin D is helpful for the upregulation of natural immune responses and the maintenance of a diverse gut microbiota [92]. Vitamin D upregulates natural immunity through the induction of antimicrobial peptides, including cathelicidin LL-37 [93,94]. The intake of a high-fat diet and vitamin D deficiency concomitantly increase the population of the hepatic pathogenic *Helicobacter hepaticus*, and the oral administration of defensin, alpha 5 (DEFA5) suppresses their population [95]. Furthermore, vitamin D has immunosuppressive effects through IL-10 production by activated regulatory T cells (Treg) cells [96]. Activated Tregs suppress various disorders caused by chronic inflammation, including asthma [97]. In addition, vitamin D has protective effects against blood vessels, is involved in anti-oxidative activity, and suppresses proinflammatory cytokine production from inflammatory cells [98,99].

## 4. Adipokine, Myokine, Cytokine

Under obese conditions, inflammation is induced by the increased production of TNF-α and the decreased production of adiponectin [100]. The suppression of GLUT4 synthesis arising from the increased production of TNF-α in muscle cells induces insulin resistance and diabetes [101]. Increased adiponectin production improves metabolic syndrome because TNF-α suppression and GLUT4 production result in the improvement of glucose intake from cells [102]. On the other hand, muscle cells produce myokines through muscle stimulation from exercise. Increased myokines promote metabolism and prevent metabolic syndrome [103]. Furthermore, cytokines are produced from a variety of cells, and TNF-α is one of the proinflammatory cytokines [104]. Figure 3 shows the schematic relationship between adipokines, myokines, and cytokines.

### 4.1. Adipokine-Producing Adipocytes

There are three types of adipocytes: white adipocytes, brown adipocytes, and beige adipocytes. White adipocytes produce adipokines, physiologically active substances produced by adipocytes. Adiponectin produced from normal white adipocytes is a protector adipokine that suppresses metabolic syndrome, which works as a longevity-related hormone. Under normal conditions, white adipocytes are small and spheroidal and produce large amounts of adiponectin. White adipocytes also produce leptin (a peptide hormone) under normal conditions. In the presence of metabolic syndrome, however, enlarged white adipocytes produce TNF-α, which is an inflammatory pathogenic contributor adipokine [105]. Under obese conditions, however, adipocytes enlarge and change their shapes and produce TNF-α, thereby inducing inflammation, insulin resistance, and diabetes. Furthermore, the production of TNF-α reduces the production of adiponectin [106]. Enlarged white adipocytes also produce angiotensinogen, heparin-binding EGF-like growth factor (HB-EGF), and plasminogen activator inhibitor-1 (PAI-1).

White adipocytes/adipose tissue (WAT) exists in the subcutis and visceral fat, where it is involved in energy storage and release. White adipocytes change their color and function to become brown adipocytes. Only small numbers of brown adipocytes/adipose tissue (BAT) exist in the interscapulum and perirenal and periaortic abdominal aorta, where they participate in heat-producing metabolism. Transient receptor potential vanilloid 2 (TRPV2) is associated with this heat-producing mechanism in brown adipocytes [107]. Brown adipocytes decrease in the ageing process. On the other hand, beige adipocytes consume fatty acids and sugars as energy sources. Therefore, increasing the number of beige adipocytes is a good strategy for preventing metabolic syndrome [108,109].

### 4.2. Myokines Released from Skeletal Muscle

Myokines are produced from muscle cells in accordance with the contraction and extension of muscles during exercise. This substance has some useful properties for preventing metabolic syndrome. Currently known myokines include interleukin-6 (IL-6), fibroblast growth factor 21 (FGF-21), secreted protein acidic and rich in cysteine (SPARC), irisin, brain-derived neurotrophic factor (BDNF), and insulin-like growth factor 1 (IGF-1). IL-6 is a useful myokine for promoting metabolism and preventing obesity and diabetes. FGF21 participates in lipid, glucose, and energy metabolism [110]. SPARC is a newly identified myokine and is related to colon cancer suppression [111]. Skeletal muscle produces irisin which correlates with the browning of white adipocytes [112]. BDNF is a neurotrophic factor that promotes brain development. IGF-1 carries glucose, as does insulin, and activates nerve cells [113]. Building muscle is an effective way to increase myokine production, which prevents metabolic syndrome and ageing. A diet rich in the amino acid leucine is recommended for increasing myokine production [114].

The activation of metabolism through exercise is needed to prevent metabolic syndrome. The number of mitochondria is increased as a result of exercise, and metabolism is improved because muscle movement requires substantial amounts of ATP [115,116]. Adipose tissue releases free fatty acid (FFA) after enlargement, but muscle cells directly consume FFA as a source of energy, decreasing FFA in the body and muscle [117]. Without exercise, however, FFA accumulates in the liver as fat and induces metabolic syndrome and other chronic inflammation-related disorders [118]. Recent research shows that quercetin supplementation in combination with strength training improves the quality of muscle [119].

Mitochondria are important cytoplasmic organelles in cells that produce energy. As muscle requires a tremendous amount of energy, the role of mitochondria in muscle cells is very important. The activation of mitochondria is necessary to maintain a healthy condition because mitochondria produce ATP, an important energy source in the body. In addition to ageing, unhealthy lifestyle habits such as overeating and a lack of exercise decrease the number of mitochondria and progress the ageing of mitochondria. Aged mitochondria produce large amounts of reactive oxide. In this context, autophagy removes aged mitochondria and abnormal proteins from cells [120]. Epigallocatechin-gallate, (EGCG) a phytochemical, induces autophagy by inducing reactive oxide species in vitro [121]. To avoid metabolic syndrome, it is important not to reduce the number of mitochondria and to activate them instead. Research shows that 6 continuous weeks of dietary habits resulting in a 25% reduction of calories increases the number of mitochondria in the muscle [122].

### 4.3. Cytokines Produced by a Variety of Cells in the Body

Cytokine is the generic name for physiologically active substances composed of small molecular proteins produced by cells. The classification of cytokines varies because of the variety of cells producing them. Adipokines and myokines, which were mentioned previously, are cytokines in a broad sense. This section focusses mainly on type 1 proinflammatory and type 2 anti-inflammatory cytokines, cytokines produced by immune cells, and cytokines related to metabolic syndrome. Type 1 proinflammatory cytokines include IL-1, IL-2, IL-6, IL-12, IL-17, IFN-γ, and TNF-α. These cytokines are produced mainly by CD4^+^ type 1 helper T cells (Th1), macrophages, and dendritic cells, which characterise the type 1 immune response. In particular, IL-1, IL-6, IFN-γ, and TNF-α are regarded as important proinflammatory cytokines. These cytokines transmit signals via the type 1 cytokine receptor (CCR1), unlike other proinflammatory cytokines [123]. On the other hand, type 2 anti-inflammatory cytokines suppress proinflammatory responses. Recently, several reports have documented the effects of these cytokines in the type 2 immune response. IL-10 and TGF-β, as well as IL-4, IL-5, and IL-13, are considered anti-inflammatory cytokines. Cancer immune therapy requires the activation of cellular immunity [124], and the elimination of parasites that have infected the body requires the activation of humoral immunity [125]. Therefore, the balance of these cytokines is very important for maintaining a healthy condition [126]. In addition, recent research has shown intriguing findings suggesting a correlation between an increase in anti-inflammatory cytokines and learning and memory [127].

IL-10 production by peripherally induced-regulatory T cells (pTreg) suppresses various immune responses [128]. A good gut microbiota activates Tregs and upregulates immune responses. The butyrate-producing bacteria *Faecalibacterium prausnitzii* and *Clostridium butyricum* MIYAIRI produce a short fatty acid butyric acid and activate Tregs [129,130]. Secretory IgA antibody plays important roles in the intestinal mucosa [131]. In addition, the gut microbiota is reportedly related to the production of maternal IgA in milk [132]. The production of IgA is important for mucosal immune functions and requires activation through dendritic cells in both T cell-dependent and T cell-independent pathways [133].

### 4.4. Effects of TNF-α and Adiponectin in Metabolic Syndrome

Under normal conditions, adipocytes produce adiponectin, which regulates energy consumption in the body through its actions in the hypothalamus, a critical part of the brain that controls endocrine function. This process suppresses liver glucose synthesis and promotes the burning of fat in the body. Adiponectin suppresses foam cell formation in blood vessels, vascular endothelium hyperplasia, and arteriosclerosis. Furthermore, adiponectin enhances insulin secretion from pancreatic β cells and glucose incorporation by skeletal muscle and promotes fat burning in the body [134]. On the other hand, TNF-α works as a pathogenic contributor and suppresses GLUT4 synthesis. This suppression induces insulin resistance, chronic inflammation, and metabolic syndrome. The increased production of TNF-α decreases the production of adiponectin and induces chronic inflammation [135]. To prevent metabolic syndrome, the suppression of TNF-α is important because chronic inflammation induced throughout the body leads to the onset of several disorders that in turn induce metabolic syndrome.

In metabolic syndrome, the level of TNF-α changes in accordance with the change of the gut microbiota [136]. Resveratrol contained in grapes suppresses proinflammatory factors including TNF-α and IL-17 via NF-κB modifications and changes gut microbiota. [137]. Furthermore, the deficiency of adiponectin decreases *Bacteroidetes* and stops the suppression of rhabdomyosarcoma [138]. It is known that patients undergoing cancer therapy tend to present with dysbiosis and a reduced ratio of adiponectin [139]. Thus, changes in TNF-α and adiponectin are highly associated with metabolic syndrome and the gut microbiota population.

## 5. Phytochemicals and Vitamin D Prevent Metabolic Syndrome and Improve Gut Microbiota

The consumption of phytochemicals and vitamin D with prebiotics and probiotics nurture good gut microbiota. Human gut microbiota changes with age [140], and microbiota in soil and in other creatures are related to microbiota in humans. Recently, these relationships of microbiota have been collectively called the microbiome and have been identified as important factors for understanding healthy longevity and the onset of several diseases in humans [141].

Athletes tend to have more gut microbiota diversity than other groups. They consume a greater variety of foods and perform much more exercise than other groups of people. This fact indicates that one’s gut microbiota is influenced by one’s diet [142]. In Japan, meat consumption has been increasing for several decades, whereas rice consumption has gradually decreased. A lack of exercise and excess caloric intake leads to obesity and fatty liver, which has a strong association with metabolic syndrome. In addition, reduced consumption of vegetables and a high salt intake result in imbalanced nutrition. Fast foods and sweetened diets are favorite choices of many people. On the other hand, the consumption of flavonoid-rich vegetables and fruits has decreased. A cohort study conducted in Japan revealed a relationship between the higher consumption of fruits and vegetables and a lower risk of death [143].

The typical Western diet, which is characterized by high fat and high sugar consumption, induces metabolic syndrome [144]. A large number of Gram-positive bacteria in the body are related to the cause of cellular senescence and liver cancer onset, and the consumption of a high-fat diet changes the gut microbiota [145]. The excess consumption of sweeteners induces dysbiosis [146]. Dysbiosis induced by high-fructose consumption is responsible for the induction of metabolic syndrome [147]. Recently, environmental factors such as endocrine-disrupting chemicals have been reported to induce inflammatory intestinal disorders, dysbiosis, and immunological disorders [148,149]. These factors are also related to obesity [150]. The intake of phytochemicals and vitamin D is associated with diverse and healthy gut microbiota, which prevents metabolic syndrome and maintains a healthy lifestyle.

### 5.1. Gut Microbiota in Obese Type and Lean Type

*Firmicutes* bacteria are usually regarded as obese-type bacteria, whereas *Bacteroidetes* bacteria are regarded as lean-type bacteria [151]. Some obese-type gut microbiota produces enzymes to digest insoluble dietary fibers such as cellulose. Some decompose cellulose into glucose, inducing obesity. Reducing the F/B (*Firmicutes* vs. *Bacteroidetes*) ratio in gut microbiota is a good strategy for preventing obesity and metabolic syndrome [152]. For this purpose, reducing sugar consumption, changing the type of sugar consumed, considering the types of ingredients that are eaten together, and changing the timing of meals (e.g., avoiding eating dinner immediately before sleep as food is not digested during sleep), are good ideas. Figure 4 shows the influence of the consumption of phytochemicals and vitamin D in gut microbiota.

*Firmicutes* are Gram-positive bacteria and the most dominant type of gut microbiota. The phylum of *Firmicutes* is subdivided into four classes: *Clostridia*, *Bacilli*, *Erysipelotrichi*, and *Negativicutes* [153]. On the other hand, *Bacteroidetes* are Gram-negative bacteria that belong to the phylum of *Bacteroidetes*; *Bacteroidetes* are also a dominant type of gut microbiota. Decreasing *Bacteroidetes* bacteria in the gut causes dysbiosis and induces a thinner mucin layer and weakened cellular adhesion in the gut, leading to leaky gut syndrome; these effects can, in turn, lead to obesity, hypertension, diabetes, dyslipidemia, arteriosclerosis, NASH, and lifestyle-related diseases [154]. On the other hand, a good microbiota consists of bacteria that result in a low F/B ratio, the lactic acid bacteria *Bifidobacterium* [155,156], and butyrate-producing bacteria. Typical butyrate-producing bacteria include non-spore-bearing *Faecalibacterium prausnitzii* and spore-bearing *Clostridium butyricum* [157].

Maintaining a well-balanced gut microbiota is important for maintaining a healthy condition, and a gut microbiota survey of newborns to long-lived people in Japan showed an intriguing relationship between longevity and gut microbiota [158]. The colonization of *Akkermansia muciniphila* in the gut is associated with balanced gut immune responses [159]. Indole-3 propionate produced by gut microbiota promotes neural regeneration and restoration [160]. In addition, sleep time and the sleep rhythm influence the gut microbiota status. Metabolic syndrome disrupts the circadian rhythms, as some gut microbes produce sleep-related hormones such as melatonin. Dysbiosis reportedly induces a lack of sleep and insomnia caused by a lack of melatonin [161]. Healthy diets such as the Mediterranean diet are recommended because healthy dietary habits prevent metabolic syndrome and nurture good gut microbiota [162].

### 5.2. Influence of Phytochemicals and Vitamin D in Gut Microbiota

Prebiotics, probiotics [163], phytochemicals, and vitamin D (3PD) are important for maintaining healthy gut microbiota [164]. Prebiotics such as water-soluble dietary fibers, dietary fibers, and microbiota-accessible carbohydrates (MAC) nurture good gut microbiota [165]. Probiotics are beneficial bacteria for healthy gut conditions and include lactic acid bacteria and *Bifidobacterium*. Phytochemicals includes terpenoids, carotenoids, and flavonoids. Vitamin D is also important for nurturing good and diverse gut microbiota. Prebiotics and probiotics are effective for the recovery of the liver from conditions such as NAFLD and NASH [166]. Table 4 summarizes the influence of phytochemicals and vitamin D in gut microbiota.

In addition, the intake of both phytochemicals and vitamin D nurtures good gut microbiota. The effects of polyphenols such as quercetin on gut microbiota have been well reported. Gut microbiota metabolize these polyphenols, and the polyphenols change the components of gut microbiota [167]. The combined administration of *Akkermansia muciniphila* and quercetin established the colonization of *A. muciniphila* in mouse gut and changed the gut microbiota composition, ameliorating obesity and NAFLD by affecting bile acid metabolism [168]. Furthermore, quercetin suppressed obesity in mice that were administered sodium glutamate. In this model, quercetin ameliorated glutamate-induced hypothalamus injury and suppressed the retinol saturase (RetSat) levels that were induced by changing the composition of gut microbiota. Furthermore, quercetin administration increased the *Bacteroides* population and corrected the F/B ratio [169]. Other experiments show the synergy of quercetin and the water-soluble dietary fiber inulin, which led to massive bodyweight loss and the amelioration of metabolic syndrome in a high-fat-fed mouse model. The dietary fiber in inulin was thought to be decomposed into short-chain fatty acids and used as energy for intestinal endothelial cells, leading to fat decomposition and an improvement in insulin resistance. Furthermore, an increased population of the lactic acid-producing *Faecalibaculum rodentium*, a reduced F/B ratio, and increased GULT4 expression were also observed [170].

Vitamin D deficiency changes the balance of microbiota in the gut because of the strong relationship between vitamin D and gut microbiota [171]. The administration of vitamin D reportedly increased the *Akkermansia* and *Faecalibacterium* populations in a study examining multiple sclerosis [172]. In addition, a lack of vitamin D can cause dysbiosis and can result in some allergic responses. Vitamin D deficiency affects the gut microbiome by impairing both gut microbiota composition and the integrity of the gut epithelial barrier. In addition, vitamin D participates in immune responses via vitamin D receptor (VDR) signaling pathways [173]. Vitamin D modulates T cells and Paneth cells through VDR to modulate antimicrobial peptide release, which is involved in interactions between host and gut microbiota. Bacterial metabolites, including butyric acid, upregulate this VDR signaling pathway [174]. Because of these mechanisms, vitamin D and gut microbiota have a strong relationship. The intake of both phytochemicals and vitamin D participates in maintaining the diversity of gut microbiota, and the nurturing of good microbiota suppresses TNF-α production.

### 5.3. Effects of Phytochemicals and Vitamin D on the Suppression of Chronic Inflammation

Recently, the anti-inflammatory effects of phytochemicals and vitamin D have been a focus of interest, along with other effects including anti-oxidation processes and enzymatic reactions. In particular, the role of flavonoids in anti-inflammatory effects is considerable. The flavonoid quercetin suppresses metabolic syndrome, improves the condition of the gut microbiota in the presence of a high-fat diet [175], and prevents hyperlipidemia [176]. Quercetin administration also alleviates NASH, a liver disorder induced by chronic inflammation through LPS and FFA [177].

Quercetin is degraded from quercetin-glycoside by the gut microbiota and is absorbed in the gut in the same manner as other polyphenols. Then, quercetin forms quercetin-glucuronide and circulates in the blood. Macrophages incorporate oxidized-LDL and transform into TNF-α producing foam cells. These macrophages produce glucuronidase and decompose quercetin-glucuronide into its activated form. Activated quercetin overturns the states of macrophages and stops the production of TNF-α. The suppression of TNF-α by the flavonoid quercetin stops the induction of chronic inflammation [178]. In addition, a variety of immune cells in the gut express vitamin D receptor (VDR) and play a pivotal role in the interaction between gut microbiota and immune cells. Vitamin D works as a brake for excessive immune response and protects its host [179,180,181].

### 5.4. Molecular Mechanism for the Prevention of Metabolic Syndrome Requiring the Suppression of TNF-α and Chronic Inflammation

Metabolic syndrome, dyslipidemia, and diabetes induce continuous TNF-α production, inducing chronic inflammation in the blood vessels and the onset of atherosclerosis. Flavonoids suppress TNF-α production from macrophages [17]. Specifically, the flavonoids quercetin, hesperidin, and naringin in fruits and vegetables suppress TNF-α production by suppressing the expression of toll-like receptor 4 (TLR4) on macrophages. Flavonoids improve dysbiosis and suppress bacterial growth related to dyslipidemia, diabetes, and metabolic syndrome [182]. Fermented grape foods from Koshu, a Japanese grape strain (named K-FGF), contain the grape skin and seed paste of *Vitis vinifera* Koshu fermented with vegetable lactic acid bacteria. K-FGF suppresses TNF-α production and prevents chronic inflammation-induced disorders, including metabolic syndrome [183,184].

The molecular mechanism for the downregulation of macrophages related to chronic inflammation is hypothesized below (Figure 5). The flavonoid quercetin suppresses membrane fluidity and lipid raft formation on the cell surface membrane [185]. TLR4 expression on the cell surface is increased in cases with chronic inflammation. However, the suppression of TLR4 downregulates cellular signal transduction [84]. Next, activated vitamin D, 1α, 25-(OH)2-D3, binds to VDR and is transported into the nucleus. Finally, the complex of vitamin D-VDR binds to the TNF-α promoter gene and stops the production of TNF-α. This mechanism suppresses metabolic syndrome by ameliorating a variety of chronic inflammation-related disorders [15]. The consumption of citrus and grape phytochemicals together with vitamin D improves gut microbiota, suppresses chronic inflammation, and upregulates immune responses. In addition, enhanced production of LL-37 by vitamin D intake helps to suppress TNF-α production via the transcription factor NF-κB pathway, which is activated by TLR4 stimulation [186].

## 6. Conclusions

Metabolic syndrome is a comorbid condition associated with a variety of lifestyle-related disorders. Because this condition is induced by several diseases including obesity, diabetes, cardiovascular disorders, liver injury, and brain disorders, the prevention of metabolic syndrome is important for maintaining a healthy lifestyle and longevity. For this purpose, nurturing healthy gut microbiota, upregulating immune responses, and suppressing chronic inflammation is important. In addition, a healthy diet and moderate exercise are recommended. The intake of phytochemicals and vitamin D can help to achieve this goal. As discussed above, the consumption of phytochemicals and vitamin D is important to maintain a healthy life.

## Figures and Tables

**Figure 1 ijms-24-02627-f001:**
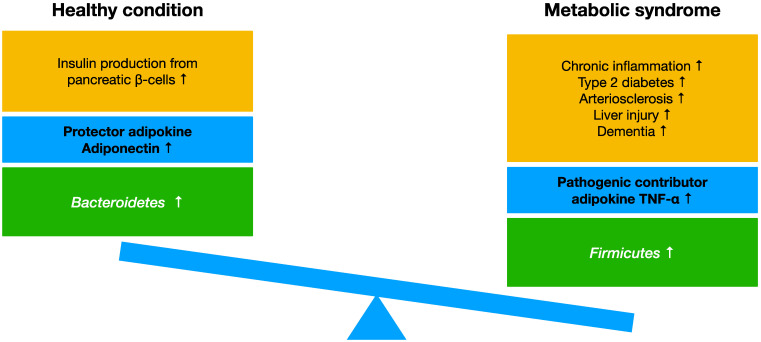
The mechanism of onset and suppression of metabolic syndrome. High-fat, high-glucose and typical Western diet causes accumulation of excess energy in the body, reduction of protective adipokine adiponectin, and increase of pathogenic contributor adipokine TNF-α induces chronic inflammation and metabolic syndrome. Metabolic syndrome induces type 2 diabetes, arteriosclerosis, and dementia. In metabolic syndrome, gut microbiota induces dysbiosis. Increased *Firmicutes* and decreased *Bacteroidetes* in the gut microbiota induce chronic inflammation and metabolic-syndrome-related disorders. Healthy diet and exercise prevent obesity, maintain good gut microbiota, and suppress metabolic syndrome.

**Figure 2 ijms-24-02627-f002:**
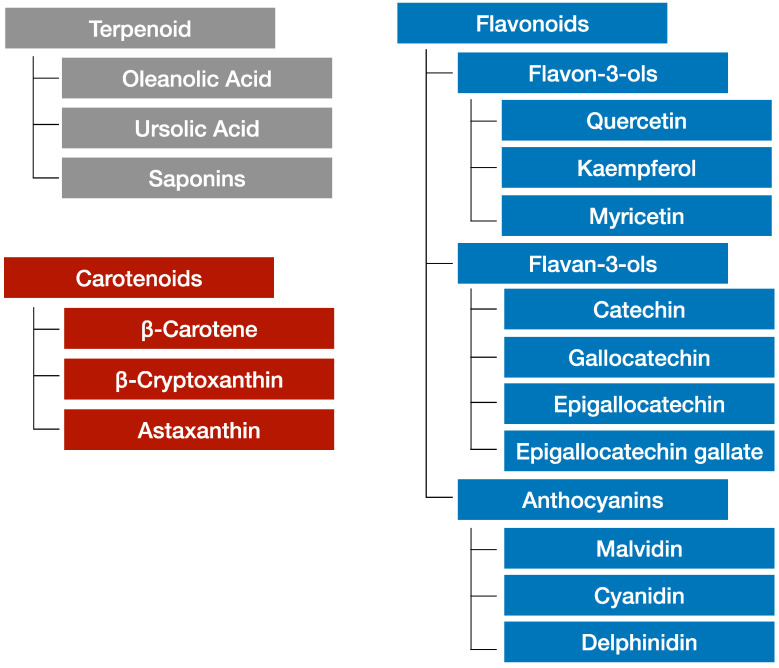
Schematic view of grape phytochemicals in anti-metabolic syndrome and anti-inflammatory effects. This figure shows each grape phytochemical contained in grapes. Grape phytochemicals are classified into terpenoids, carotenoids, and flavonoids. Grape terpenoids are subdivided into oleanolic acid, ursolic acid, and saponins. Grape carotenoids are subdivided into β-carotene, β-cryptoxanthin, and astaxanthin. Grape flavonoids are subdivided into flavon-3-ols, flavan-3-ols, and anthocyanins.

**Figure 3 ijms-24-02627-f003:**
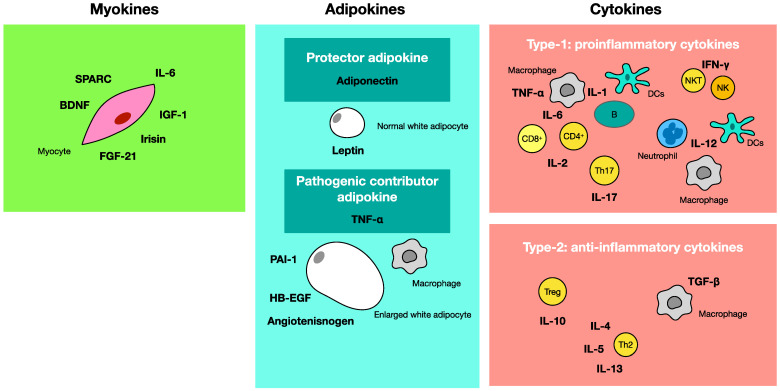
Schematic relationship of adipokines, myokines, and cytokines. This figure shows adipokines, myokines, and cytokines and their producing cells. Adipokines include protective adiponectin produced from normal white adipocytes, and pathogenic contributor TNF-α which is produced from enlarged white adipocytes and macrophages. Myokines are produced by muscle cells. Cytokines are physiologically active substances produced by cells in the body. This scheme shows immune regulatory cytokines, namely Type 1 proinflammatory cytokines and Type 2 anti-inflammatory cytokines. These cytokines are produced from macrophages, dendritic cells, neutrophils, NK cells, NKT cells, T cells, and B cells. CD4^+^ T cells are classified into Th1/Th2, Th17, and regulatory T cells (Treg), and each of them produces different cytokines. TNF-α belongs to both adipokines and proinflammatory cytokines. IL-6 is also classified into proinflammatory cytokines and myokines.

**Figure 4 ijms-24-02627-f004:**
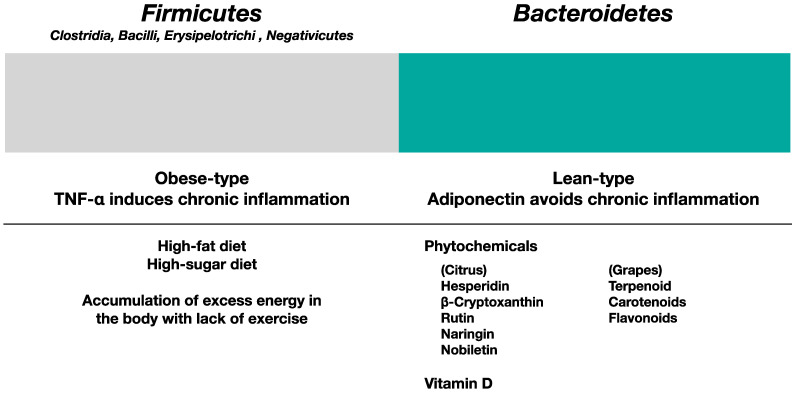
The influence of the consumption of phytochemicals and vitamin D for the gut microbiota. High-fat, high-sugar, high-calorie diet is responsible for obesity; these conditions lead to metabolic syndrome due to the lack of exercise. This status causes chronic inflammation induced by pathogenic contributor TNF-α, and the state of microbiota becomes *Firmicutes* dominant. On the other hand, the intake of phytochemicals and vitamin D preferentially induces beneficial gut microbiota with increasing protector adiponectin. Mainly the increase of *Bacteroidetes* and the decrease of the F/B ratio is beneficial for maintaining a healthy condition.

**Figure 5 ijms-24-02627-f005:**
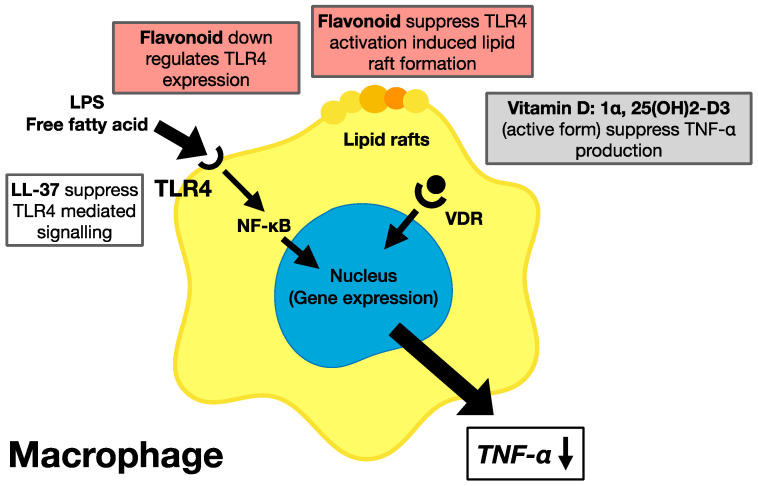
Molecular mechanism for the suppression of TNF-α and chronic inflammation. This figure shows molecular mechanism of TNF-α suppression and chronic inflammation by phytochemicals, flavonoid, and vitamin D. TNF-α induces chronic inflammation and related disorders and finally induces metabolic syndrome. Flavonoid quercetin reduces the expression of toll-like receptor 4 (TLR4) on the surface of macrophage and suppresses the formation of lipid raft on the cell membrane. Suppression of TLR4 stops activation of the NF-κB signalling pathway. In addition, vitamin D-enhanced LL-37 also suppress the activation of transcription factor NF-κB via TLR4 stimulation. Then, the active form of vitamin D, 1α, 25(OH)2-D3, conjugates with the vitamin D receptor (VDR) in the cytoplasm and is translocated to the nucleus. The vitamin D-VDR complex attaches to the gene promoter region of TNF-α and stops production of TNF-α. Interaction between phytochemicals and vitamin D suppress the production of TNF-α and induction of chronic inflammation. Finally, lowered TNF-α production suppresses the chronic-inflammation-related disorders including diabetes, cardiovascular diseases, and Alzheimer’s and suppresses metabolic syndrome.

**Table 2 ijms-24-02627-t002:** Anti-metabolic syndrome and anti-inflammatory effects of grape phytochemicals.

Grape Phytochemicals	Effects	Subjects	Ref
Wine	Health-improving effects of wine	Human	[74]
Resveratrol	Maintaining health condition in high calorie intake	Mouse/ *D. melanogaster*/ *C. elegans*/ *S. cerevisiae*	[75]
	Reducing risks of NAFLD and gut dysbiosis	Mouse	[76]
	Atherosclerosis prevention	Human	[83]
Red wine polyphenol	Atherosclerosis prevention, effects in vascular smooth muscle cells	Human and Bovine endothelial cells	[77]
Red wine bioactive compound	Anti-oxidative, thrombin inhibition, lipase inhibition	Cells/Activity screening kit	[78]
Flavonoids	Anti-oxidative, anti-inflammatory, anti-carcinogenesis, circulatory system disease prevention	Human	[79]
	Alleviating collagen-induced arthritis	Mouse	[84]
	Anti-ageing	Human	[85]
Grape phytochemicals, GSE, K-FGF	Alleviating intestine related disordered	Human/ Rat/Mouse	[80]
Grape seed extract (GSE)	Lung fibrosis prevention	Mouse	[81]
Grape seed flan-3-ols	Analysis of biosynthetic pathways in nutraceuticals	Physical analysis	[82]
Procyanidin	Preventing senescence	Mouse/ Human cells	[86]

**Table 3 ijms-24-02627-t003:** Anti-metabolic syndrome and anti-inflammatory effects of vitamin D.

Effects	Subjects	Ref
Boosting natural immunity, maintaining diversity of gut microbiota	Human	[92]
Antibacterial peptide LL-37 induction, upregulation of innate immunity	Human	[93]
Strengthen natural immunity by the induction of antibacterial peptide	Human	[94]
Gut microbiota modification, insulin-resistance, NAFLD by defensins	Mouse	[95]
Treg activation by IL-10 production, suppression of inflammatory immune response	Human/ Human cells	[96]
Suppression of chronic inflammation related disorders by Treg activation	Mouse	[97]
Vascular vessel protection, anti-oxidative, proinflammatory cytokine suppression	Human	[98]
Upregulation of immunity against COVID-19 infection	Human	[99]

**Table 4 ijms-24-02627-t004:** The influence of phytochemicals and vitamin D in gut microbiota.

Source	Effects	Subjects	Ref
Polyphenols	Microbiota in metabolic disorders	Human/ Rat/Mouse	[167]
Quercetin	Improvement of obesity and NAFLD	Mouse	[168]
	Correct F/B ratio, obesity	Mouse	[169]
	Insulin resistance, increases *Faecalibaculum rodentium,* improves F/B ratio, increases GULT4	Mouse	[170]
Vitamin D	Gut microbiota modification	Human	[171]
	Increasing *Akkermansia* and *Faecalibacterium* (in multiple sclerosis)	Human	[172]
	Improvement of gut dysbiosis	Human	[173]
	Antimicrobial peptide release, gut microbiota interaction	Human/Mouse	[174]

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
