# Peer review of "Prevention of Metabolic Syndrome by Phytochemicals and Vitamin D"

_ijms, 2023, doi:10.3390/ijms24032627_

Round 1
Reviewer 1 Report (New Reviewer)
Dear authors,
The manuscript presents an interesting and comprehensive paper on the effect of phytochemicals and vitamin D on metabolic syndrome. In this way, the first question refers to the fact that most of the work focuses on describing the pathologies and only subheading 5 focuses on describing the effects of the consumption of these compounds. For this reason it is recommended to either summarize the rest of the subheading, or preferably try to augment this section.
Figure 3 should be modified to be more illustrative and better explain the content, illustrating relationships more adequately.
Subheading 4 should be linked to the next subheading which is the core of the job.
Figure 4 must be cited before appearing in the text
It is recommended to include a figure on the effect of the consumption of phytochemicals and vitamin D on the intestinal microbiota and how these communities vary.
Best regards
Author Response
Dear Reviewer 1,
Thank you very much for your review. We are really pleased to see your score was very high (21/25).
We will send a new cover letter to you. We found your comment in reviewer 1 at the Journal website.
Please see the attachment.
We really appreciate your comment, we are looking forward to hearing from you soon.
Best regards,
Kazuki Santa
Tokyo College of Biology

Reviewer 2 Report (New Reviewer)
A review article by Kazuki Santo, Yoshio Kumazawa, and Isao Nagaoka deals with the prevention of metabolic syndrome with phytochemicals and vitamin D, a very interesting and current topic.
In the first part, the authors list the phytochemicals that are mainly found in citrus fruits and grapes and explain their positive effects in the prevention of metabolic syndrome. In the second part of the article, an explanation of the pathophysiological mechanisms of the metabolic syndrome and the role of certain cytokines in its development is given, along with an explanation of the effects of phytochemicals and vitamin D in its prevention.
The article is written very clearly but is very basic and sounds most like a student review article. It is unnecessarily long with a lot of repetition. It is divided into several smaller chapters, in which most of the information is repeated. The paper cited a large number of references (mostly basic) and sounds most like a student review article. It is unnecessarily long with a lot of repetition. It is divided into several smaller chapters, in which most of the information is repeated. The paper cited a large number of references (179), and the references are relevant and present the most recent publications, but the authors did not provide a good synthesis of the literature. A better summary of data from the literature would ensure the avoidance of repetition, shorten the article, and be beneficial.
The review is easy to understand, although there are a few uncertainties that were probably generated by poor translation, but it is not written as I would expect to see in an academic journal article. In an article that should be published in a scientific journal, adjectives like "good" and "bad" cytokine are not suitable as they make the article sound unscientific.
Four simple tables with uncommon information, including the type of paper and reference number, are also included in the work. The tables below don't need to be explained in great length because they are clear and are explained in the text. On the other hand, it is required to define the abbreviations that were used below the table. The article also includes four basic figures, each of which is unnecessarily described in detail below it and later in the text.
Pay close attention to the following, please:
Line 115: "Diabetes is one of the diseases causing metabolic syndrome" (not that it causes, but it is a part of metabolic syndrome).
Line 117: not necessarily low insulin secretion but rather inadequate
Line 124: It is not clear what the authors wanted to say with this sentence, "This state is caused by the rapid decomposition of starches into glucose after meals by the secretion of insulin." Please rephrase it.
Line 138: "Arteriosclerosis is a typical cardiovascular disease that causes metabolic syndrome." This statement is incorrect; the probable reason is an inadequate translation.
Line 163: "LPS" please explain the abbreviation
Line 340, "DEF A5", please explain the abbreviation.
Line 343: "Treg," please explain the abbreviation
Line 356-357: unnecessary sentence, repetition
Line 357, "the most popular cytokines," does not sound scientific and should be rephrased.
Line 369 "HB-EGF and PAI 1" - please explain the abbreviations
Line 402-403: Please provide explanations for all abbreviations in the order they appear.
Line 526 - is it „decreasing Firmicutes bacteria induces dysbiosis,,,," that you wanted to say?
Line 635: It is not clear what the abbreviation K-FGF means.
TRL4 abbreviation is explained in line 642 but it was used several times before
Author Response
Dear Reviewer 2,
Thank you very much for your report. We are pleased to see your comment you will sign your review report. Furthermore, we really appreciate that you evaluate this manuscript as very interesting and current topic.
We will send a new cover letter to you. We found your comment in reviewer 2 at the Journal website.
Please see the attachment.
If you have any other comments that our manuscript needs to be corrected please do not hesitate to tell us.
Once again, thank you very much for your respectful comments.
We are looking forward to hearing from you at your earliest convince.
With kind regards,
Kazuki Santa
Tokyo College of Biology

Round 2
Reviewer 2 Report (New Reviewer)
The manuscript has been sufficiently improved to warrant publication in IJMS.
Minor suggestions
Line 144 - The sentence should be corrected. Maybe : "Glucose metabolism disorder, hypertension, and dyslipidemia induce the onset of atherosclerosis."
Line 396 - " produced by adipocytes" -you have already said at the beginning of the sentence.
Line 399 - It is not clear what did you want to say with this sentence. Adiponectin is an adipokine, leptin is a hormone.
Author Response
Dear Reviewer 2,
Thank you very much for your comments. We are pleased to see your comment that the manuscript has been sufficiently improved to warrant publication in IJMS.
We made minor changes in accordance with your report.
Please see the attachment.
Once again, thank you very much for your review. We are really pleased that our manuscript has been checked by you.
Looking forward to hearing from you at your earliest convenience.
Yours sincerely,

This manuscript is a resubmission of an earlier submission. The following is a list of the peer review reports and author responses from that submission.
Round 1
Reviewer 1 Report
This literature review by Kazuki Santa deals with the effects on inflammation and, by extension, the Metabolic Syndrome of some compounds with bio-active activity. The article it is about a very interesting but over-seen topic and does not add new content to the scientific literature. The topics covered are superficial and no new information or new conclusions are integrated. In addition, several aspects should be considerably improved in order to be valued for potential publication. Here are some issues that, in my opinion, must be considered by the author:
· -The wording of the article could be considerably improved. Throughout the text, imperative statements are used, such as “Prevention of metabolic syndrome is important because this symptom causes diabetes, cardiovascular disease, and liver disorders,” when metabolic syndrome is far from being a symptom. Or, “In addition, beige adipocytes avoid causing metabolic syndrome because”, when beige adipocytes DO NOT AVOID by themselves! Many similar phrases can be found throughout the text.
· -Lack of rigor when describing terms. For example: “Metabolic syndrome is relatively new disease” (it is neither a symptom nor a new disease, it is a set of metabolic conditions/alterations). Another example: “Diabetes is the first symptom induced by metabolic syndrome related with obesity and divided into type 1 and type 2 diabetes". This phrase is totally wrong... Is diabetes a certain consequence of the metabolic syndrome? What does type 1 diabetes paint here, when it has a totally different etiology? You can find many phrases similar to throughout the text.
· -Figures are inconsistent with each other in relation to format. Furthermore, they contain errors: Fig. 1 INFα should be TNFα? The information that is intended to be presented in table 1 has a similar purpose to Figure 2. Why is it chosen to present one in the form of a table and the other in the form of a figure? Why don't bibliographic citations appear in Figure 1?
· -Lack of coherence. The article explains the benefits of a set of phytochemicals, including those found in citrus. However, later Vitamin D is also added (absent or very little present in these foods). Why this decision? There is a lot of literature on Vitamin D and Metabolic Syndrome and this review does not add anything new. It is also not understood why the French paradox is not mentioned but Resveratrol does not appear anywhere as a bio-active of interest.
· -The classification into good and bad cytokines (adipokines, myokines, ...) is very presumptuous and depends on the context. Can not generalize. To give an example, the classification of high levels of leptin as beneficial in cases of Metabolic Syndrome is, more than anything, questionable.
· -The coherence of the text is very weak. Above all, an excessive repetition of sentence structure is used that seems transcribed from a speech rather than written on purpose. Two examples: “Myokines are mainly produced from muscle cells. Myokines are produced by muscle cells stimulated by the exercise.” And “Mitochondria is the place to produce energy. Mitochondria is one of the cell organ-339 nelle, producing ATP which is necessary for all the living creatures”.
In brief, this review needs important improvements to assess its potential publication in high-impact journals.
Author Response
Dear Sir/Madam,
Thank you very much for your review. I saw your reviewer comments at this webpage. Your comment was tougher than another reviewer comment. What I can do is doing my best for significant revision to resubmit this manuscript in IJMS. Please give me additional time for this manuscript.
Kind regards,
Reviewer 2 Report
This manuscript introduced representative metabolic disorders including diabetes, cardiovascular diseases, liver disorders, and brain disorders. Then, the author summarized the preventive effects of phytochemicals from citrus and grapes and vitamin D on metabolic syndrome. Adipokine, myokine, cytokine from different tissues or organs were also described. Finally, this article describes their role in alleviating metabolic-related diseases by improving the state of gut microbiota, up regulating immune responses and suppressing chronic inflammation. However, the purpose of this study is not clear by listing phytochemicals and vitamin D together. Do they have mixed effects or functions on metabolic diseases? Metabolic diseases, phytochemicals, vitamin D, adipokines are indeed associated with each other, but the author failed to focus on the phytochemical and vitamin D intervention on metabolic diseases alone, rather there were a variety of dietary interventions introduced in the context which made it difficult to understand the purpose of this study.
Other comments:
1、 The description of EFFECTS of phytochemicals in Table 1 is not clear. Some are related diseases, some are biological functions, and some are effects on specific diseases. Also, some of those are marked as“mouse”or“Clinical Trial”but some are not. The text on page 6 can be incorporated into the table.
2、 In page 13, INFLUENCE OF PHYTOCHEMICALS AND VITAMIN D IN GUT MICROBIOTA,the anti-inflammatory mechanisms of quercetin and vitamin D rather than their influence in gut microbiota seem to be the focus here. Also, more references should be supplemented here.
